# Current Status and Management of Chronic Myeloid Leukemia in the Gulf Region: Survey Results and Expert Opinion

**DOI:** 10.3390/cancers16112114

**Published:** 2024-05-31

**Authors:** Giuseppe Saglio, Mohamed Yassin, Ahmad Alhuraiji, Amar Lal, Arif Alam, Faraz Khan, Fatima Khadada, Hani Osman, Islam Elkonaissi, Mahmoud Marashi, Mohamed Abuhaleeqa, Murtadha Al-Khabori, Ramesh Pandita, Salam Al-Kindi, Shakir Bahzad, Dayane Daou, Yasmin Al Qudah

**Affiliations:** 1Department of Hematology, University of Turin, 10124 Torino, Italy; giuseppe.saglio@unito.it; 2Hamad Medical Corporation, Doha P.O. Box 3050, Qatar; 3Kuwait Cancer Control Center, Shuwaikh 1031, Kuwait; aalhuraiji@gmail.com (A.A.); ramesh.pandita@gmail.com (R.P.); shakirbahzad@hotmail.com (S.B.); 4Tawam Hospital, Al Ain P.O. Box 5674, United Arab Emiratesarifalam@hotmail.com (A.A.); haniyousif99@gmail.com (H.O.); 5American Hospital, Dubai P.O. Box 3050, United Arab Emirates; fkhan@ahdubai.com; 6Shaikh Shakhbout Medical City, Abu Dhabi P.O. Box 11001, United Arab Emirates; i.konaissi@gmail.com; 7Dubai Hospital, Dubai P.O. Box 7272, United Arab Emirates; marashi3m@yahoo.com; 8Yas Clinic, Abu Dhabi P.O. Box 44852, United Arab Emirates; mohamed.haleeqa@yasclinicgroup.ae; 9Department of Hematology, Sultan Qaboos University, Muscat 123, Oman; khabori@squ.edu.om (M.A.-K.); sskindi@yahoo.com (S.A.-K.); 10Gulf, Novartis Pharma Services AG, 4056 Basel, Switzerland; dayane.daou@novartis.com; 11Oncology, Gulf, Novartis Pharma Services AG, 4056 Basel, Switzerland; yasmin.al_qudah@novartis.com

**Keywords:** chronic myeloid leukemia, Gulf Council countries, survey, expert opinion

## Abstract

**Simple Summary:**

There is a noticeable dearth of information regarding chronic myeloid leukemia (CML) within the Gulf region. This study seeks to address this gap by offering comprehensive insights into the status of CML in the Gulf, encompassing aspects such as diagnosis, testing, treatment objectives, toxicities, and discontinuation. Through a survey completed by 15 experts and subsequent discussions, this research sheds light on the management and treatment approaches for CML within the region. Remarkably, the practices observed among these experts closely align with global standards, although unique challenges pertinent to the Gulf region were distinctly identified during the discourse.

**Abstract:**

Studies on chronic myeloid leukemia (CML) in the Gulf region are scarce, consisting of a survey and expert meeting that included 15 experts in 2023 which discussed CML diagnosis, testing, treatment objectives, toxicities, and discontinuation in the Gulf region. Most patients were reported to be in first-line therapy, and the most common treatments were imatinib/imatinib generic in first-line and dasatinib in second- and third-lines. Mutation analysis was not reported to be routinely performed at the time of diagnosis but rather in case of progression to accelerated/blast phase or any sign of loss of response. While all participants were aware that BCR-ABL should be monitored every three months during the first year of treatment, 10% reported monitoring BCR-ABL every six months in practice due to test cost and lab capability. The most important first-line therapy objective was “achievement of major molecular response” (MMR) in younger patients and “overall survival” in older ones. The most important treatment objectives were “MMR” and “early molecular response followed by prolongation of overall survival” in the short term and “treatment-free remission” in the long term. The current practices in CML in the Gulf region appear to be similar to global figures.

## 1. Introduction

Chronic myeloid leukemia (CML) is a Philadelphia chromosome-positive myeloproliferative neoplasm characterized by the expression of BCR-ABL1, a tyrosine kinase protein [1]. A recent study on the burden of CML across 204 countries and territories has shown that globally, the incident cases of CML increased by more than 1.5-fold in the past 30 years from 42.7 × 10^3^ in 1990 to 65.8 × 10^3^ in 2019 [2].

While older CML treatment choices were limited to interferon-alpha and allogeneic bone marrow transplantation, advances in CML treatment have granted patients with chronic phase CML (CP-CML) a life expectancy similar to that of the general population [3]. Tyrosine kinase inhibitors (TKIs) emerged more than two decades ago as a new targeted therapy for CML, with the first-in-class drug Imatinib as a first-generation TKI, followed by three second-generation TKIs (2GTKI) (nilotinib, dasatinib, and bosutinib), and a third-generation TKI (ponatinib) [4]. Asciminib, a first-in-class allosteric inhibitor of BCR-ABL1 kinase activity, was recently introduced. In the ASCEMBL trial, asciminib resulted in a higher major molecular response (MMR) rate (25.5%) than bosutinib (13.2%) at week 24 in patients with CP-CML who were resistant/intolerant to ≥2 prior TKIs. The incidence of adverse events leading to treatment discontinuation was lower with asciminib [5]. Several other treatment options are currently under study, including new molecules as well as combinations of different drugs [6].

The European LeukemiaNet (ELN) [7] and National Comprehensive Cancer Network (NCCN) [8] guidelines include recommendations on CML monitoring, evaluation of response, and different treatment options in each line of therapy. Both guidelines state that first- and second-generation TKIs are all appropriate first-line treatments for patients with CP-CML, while changing to a 2GTKI is mandatory in cases of failure/resistance to the first-line TKI. After resistance to two TKIs, the ELN recommends switching therapy to another 2GTKI or to ponatinib [9].

While several studies on CML in Western populations exist, similar studies are scarce in the Middle East [10], and the Gulf Cooperation Council (GCC) countries are no exception. While a few studies in the GCC have examined CML patients’ characteristics and outcomes, most of those studies included small numbers of CML patients, either alone or as part of a sample with different hematological malignancies [11,12,13]. Areas such as CML incidence/prevalence, diagnosis, management, and treatment patterns remain largely unstudied. In the absence of similar data, there was a need for an expert meeting and survey to understand the current status of CML and management practices in the GCC region. Accordingly, experts from the region agreed to convene to close the gap. The current status and management of CML was discussed in the “CML Gulf Workshop” held in Kuwait on 6 May 2023. A panel of experts from the GCC region who are involved in the management of CML attended the workshop. During the meeting, the experts discussed several topics with the aims of (1) assessing the current care of patients with CML in the participating countries compared with international recommendations, (2) identifying local challenges faced by physicians in implementing these recommendations, and (3) developing practical solutions to support physicians in optimizing the management of CML patients in the region.

## 2. Materials and Methods

A mixed methods design was adopted: a quantitative survey was followed by a qualitative analysis through a focused group meeting where more insights were collected and analyzed to reach the final results (see Appendix A).

A multidisciplinary group of 15 specialists including hematologists, oncology clinical pharmacists, and geneticists/molecular pathologists from the United Arab Emirates (UAE), Qatar, Kuwait, and Oman were invited to a meeting organized by Novartis, one of whom served as the meeting moderator (Dr. Mohamed Yassin, Hamad Medical City, Qatar). Giuseppe Saglio (University of Turin, Italy) attended the meeting as a speaker, presenting recent guidelines for the use of treatments beyond second-line in CML. In addition, six subjects from the sponsor’s medical team and two consultants attended the meeting.

A pre-meeting survey was sent to all participants before the meeting. The survey was an adaptation of the TARGET survey that explored the real-world management of CML across 33 countries in 2017 [14]. During the meeting, the survey results were then presented and discussed in a dedicated session, during which some of the questions were rephrased for better clarity and appropriateness based on current practice in the region. After the meeting, the survey was sent to the participants again and their updated answers were collected and reported accordingly.

The survey covered several topics related to current CML practices including diagnosis, testing and standardization, treatment objectives and trends, as well as treatments’ toxicities and discontinuation.

The present paper represents the summary of the survey results and the experts’ opinions on the discussed topics.

### Statistical Analysis

All analyses were performed using SPSS v.28.0 (IBM Corp., Armonk, NY, USA). Data was summarized using descriptive statistics. Categorical variables were summarized using frequencies and percentages.

## 3. Results

### 3.1. Participants

Out of 15 participants who received the survey, 13 responded to it (87%).

Most of the participants reported having been treating CML for more than 15 years (62%), and 23%, 31%, and 23% of participants have provided care for 10–20, 20–50, and 50–100 CML patients in the previous 12 months, respectively.

Most of the participants’ patients were in first-line therapy (63%). The most common treatments were imatinib/imatinib generic (55%) in first-line and dasatinib in second line (44%) and third-line (30%) (Table 1).

### 3.2. Diagnosis of CML

The initial test performed when a patient with a clinical presentation compatible with CML diagnosis is encountered was reported to be karyotyping by all respondents, mainly conducted on bone marrow aspirates (100%) and blood samples in some cases (15%). In addition, 77% of the experts reported performing real-time quantitative polymerase chain reaction (RQ-PCR) for BCR-ABL as a baseline at the time of diagnosis, while mutation analysis was not reported to be routinely performed at the time of diagnosis of CML in their practice (Figure 1).

### 3.3. Molecular Monitoring and Mutation Analysis

The respondents reported differences in the types of conversion factor and standardization processes used in their centers. The conversion factor was reported to be established through an international reference lab in 54% of the respondents’ institutions, while more than two-thirds of participants (69%) did not know when the last standardization process was conducted in their centers’ labs.

The majority of respondents had access to deep molecular remission (DMR) testing by RQ-PCR to a level of molecular response 4 (MR4) or deeper (90%). The reported limits of detection for the RQ-PCR available at the respondents’ centers were MR4.5 (70%), MR4 (10%), MR5 (10%), and other (10%), which was not specified. When receiving a RQ-PCR lab report with BCR-ABL level as “not detectable”, 90% of respondents reported that they would be confident that the response is MR4.5 and above, while 10% believed that it was MMR.

All BCR-ABL lab reports provided information on results in International Scale (IS) (100%), while 60% included the assay type and BCR-ABL copy number, 50% included the control gene copy number and trends over time, while only 10% of reports included the raw results.

Mutation status was reported to be mainly assessed in the case of progression to accelerated or blast phase or any sign of loss of response (hematologic or cytogenetic) (90%) (Figure 2). Some participants reported never assessing mutation status, which was attributed to lack of access and financial issues.

While all participants were aware that BCR-ABL should be monitored every three months during the first year of treatment, 10% reported monitoring BCR-ABL every six months in practice. This non-adherence to the ideal testing frequency was attributed to the cost of the test (23%), lab capability (23%), or other non-specified reasons (54%).

### 3.4. Treatment Objectives

In younger patients, the most important first-line therapy objective was “achievement of MMR”, while in older patients, “overall survival” was considered the most important (Figure 3 and Figure 4).

The majority of respondents considered that MMR (70%) and “early molecular response (EMR) followed by prolongation of overall survival” (70%) were the most important short-term treatment objectives, while all respondents considered treatment-free remission (TFR) to be the most important long-term treatment objective.

The most important factor influencing treatment decision for first-line therapy was reported to be Sokal, EUTOS, or ELTS score (60%) (Figure 5).

A 2GTKI was reported to be preferred over imatinib as frontline therapy for a patient with CP-CML when TFR is a high-priority goal for the patient (100%) and for patients with high Sokal risk scores (90%).

### 3.5. Treatment-Related Toxicities

Only 9% of participants stated that they would change treatment for persistent grade 1 hematological or non-hematological toxicities. This percentage increased to 55% and 85% for Persistent Grade 2 and Grade 3/4 hematological toxicities respectively, and to 82% and 92% for Persistent Grade 2 and Grade 3/4 non-hematological toxicities, respectively (Figure 6).

### 3.6. Treatment Discontinuation

All of the respondents’ centers provide rapid intervention if the patient needs to restart therapy and rapid turnaround of RQ-PCR results (within four weeks) (100%), and 92% of the respondents’ centers have the capacity to test patients every four to six weeks when required.

## 4. Discussion and Expert Opinion

Our survey aimed to describe current practices and management of CML in the GCC region, which comprises a unique group of patients in a unique clinical setting. The percentage of patients in first-, second-, and third-lines in this survey was 63%, 26%, and 11%, respectively. Those numbers cannot be confirmed by solid figures, considering the scarcity of studies describing the CML population in the GCC region. Among regional figures, a group of experts from Turkey reported similar estimates of 59%, 27%, and 12% of CML patients receiving first-, second-, and third-line treatment, respectively [15]. The TARGET survey reported slightly higher numbers in first-line treatment (66%, 25%, and 9%, respectively) [14].

Karyotyping was reported to be the initial test performed for CML diagnosis by all respondents, while mutation analysis is not routinely performed at the time of diagnosis. This is in line with ELN recommendations that consider fluorescence in situ hybridization (FISH) and PCR as non-mandatory at diagnosis [7]. Experts also agreed that “next-generation sequencing (NGS) remains for research and exploratory purposes and is not yet a standard test for diagnosis of CML in the GCC”. In addition, the experts noted that “some mutations may not be found if sequencing techniques had low sensitivity”. The experts suggested that “the use of diagnostic procedures should consider prioritization, adaptation to patient profile, and specificity of the center approach”.

Only half of the respondents’ institutions utilize an international reference lab to establish the conversion factor, and more than two-thirds of the respondents did not know when the last standardization process was conducted at their respective institutions. These figures highlight the need for standardized reporting across the GCC to unify the understanding of test results and facilitate patient follow-up and monitoring across different institutions and sectors.

DMR testing by RQ-PCR to a level of MR4 or deeper was reported to be accessible to the majority of the experts. IS is used in all BCR-ABL lab reports, while assay type, BCR-ABL copy number, control gene copy number, and trends over time were missing in almost half of the reports. Mutation status was reported to be mainly assessed in case of progression to accelerated or blast phase or any sign of loss of response. However, lack of access and financial issues precluded some experts from ever assessing mutation status. Similar figures were suggested by the Turkish expert group and the TARGET survey, which might imply that those barriers are common across different countries in the region [14,15].

There was some disparity between the ideal and actual BCR-ABL testing frequencies, sometimes attributed to the cost of the test and lab capability.

“Achievement of MMR” and “overall survival” were considered the most important first-line therapy objectives in younger and older patients respectively. TFR was considered to be the most important long-term treatment objective. The experts suggested that “CML treatment objectives and goals are not fixed; those vary a lot with time and are affected by patient age and characteristics. Multiple subgoals are also often prioritized e.g., an older patient’s goal might be overall survival but the subgoal might be their quality of life” The ADAPT UK study has shown that despite being considered a more forgiving measurement, practitioners in the United Kingdom still prioritize MMR as a treatment goal [16]. More evidence has been emerging recently to support the adoption of TFR as a target in an effort to decrease treatment-related toxicities, improve patients’ quality of life, and decrease treatment costs [7,8]. Alam et al. reported real-world data from six patients in Tawam Hospital, UAE, who were on TKI therapy for 11 years and had undetectable levels of BCR-ABL1 transcript for more than 2 years. Following a trial of TFR, half of the patients reached the six-month milestone: two remained at undetectable levels, one patient had MR4.49, and none of the patients experienced TKI withdrawal syndrome [17]. Another study at the Kuwait Cancer Control Center analyzed data from 42 patients who had a trial of TFR. Most patients had a low-risk Sokal/ELTS score, their median age was 38 years, TKI exposure was 103 months, and DMR was 70 months. TFR rate was 74% at a median follow-up of 50 months and all patients who lost TFR were able to regain DMR upon restarting TKI [18].

Risk scoring using the Sokal, EUTOS, or ELTS scores was considered the most important factor influencing treatment decision for first-line therapy. The ELN recommends using ELTS as a prognostic evaluation system [7]. There might be a need in the near future to unify scoring in practice in the GCC region as it affects the selection of treatment in different lines of therapy and is an important measure when changing treatment as well.

In the same context, patients who consider TFR to be a high-priority goal and those with high Sokal risk scores are considered for 2GTKI rather than imatinib as frontline therapy.

The experts reported different real-world clinical experiences with TKIs. Ponatinib was reported to be superior to 2GTKIs in terms of tolerability and efficacy, however, due to its associated cardiovascular risk, it was advised that a reduced-dose schedule be followed as per the OPTIC trial [19]. Asciminib was reported to have a favorable safety and tolerability profile. The experts agreed that its efficacy in T315I mutation cases, low risk of cardiovascular events, and lack of drug–drug interactions were advantageous. It was suggested that in the future, Asciminib could have a potential role in first-line therapy and combination therapy, although the availability of data and access to this new molecule in some countries remain an issue. Olverembatinib and PEG-interferon were suggested as potential treatment options, but they warrant further research into safety and efficacy.

The experts reported that in clinical practice, they use 2GTKIs as primary therapy for young patients with proven success and that, other than the Sokal risk score, they check the depth and duration of response to therapy. If a low-risk patient had an interest in pursuing a treatment goal of TFR, they could potentially benefit from a 2GTKI as well. In real-world practice in the GCC, patients are relatively young, and only a minority of patients are older and have comorbidities that preclude the use of 2GTKIs.

Two studies from Saudi Arabia and one study from Bahrain have reported mean/median ages of patients of 40–43 years [11,12,13]. Previous studies have shown the importance of considering each patient’s comorbidities at treatment selection whether at diagnosis or in subsequent treatment lines [20]. Patients’ age and comorbidities remain the most crucial factors in TKI selection and monitoring to adjust the dose if needed [21]. It is also important to note that comorbidities are coupled with concomitant therapies leading to a high risk of potential drug–drug interactions [22].

More experts reported changing treatment in the case of Persistent Grade 2 and Grade 3/4 non-hematological toxicities as compared to Persistent Grade 2 and Grade 3/4 hematological toxicities. A previous expert consensus has suggested a potential explanation for this observation, stating that hematologists are more accustomed to handling hematological toxicities as compared to other types of toxicity [15].

The majority of the respondents’ centers have the capacity to test patients (PCR) every four to six weeks when required (e.g., after stopping treatment), however, while testing patients every four to six weeks in the first six months is the optimal practice based on the ELN guidelines, there was disagreement among the experts on the necessity for such a close monitoring frequency, since losing remission is an unavoidable risk at different points of treatment, after weeks, months, or even years.

Some topics that were not covered in the present survey and expert opinion are worth exploring in future studies, including the patient journey, access to treatments in each GCC country, CML patients’ characteristics and comorbidities, among others.

## 5. Conclusions

The current practice in CML management in the GCC region appears to be similar to the global practice. The experts considered the meeting to be an important starting point in raising issues in the management of CML in the region, hoping that it would serve as a foundation for a regional consensus on the discussed topics. It was highlighted that by continuing to contribute ideas and solutions through research and evidence-based practice, patient outcomes will improve as a result. Similar discussions shall contribute to resolving knowledge gaps on unmet needs and unanswered questions on CML in GCC and improving CML patient care through encouraging personalized management of CML in the region. A continued discussion through follow-up meetings was agreed upon among the experts, with the aim of reaching a regional consensus to be published and adopted in clinical practice.

## Figures and Tables

**Figure 1 cancers-16-02114-f001:**
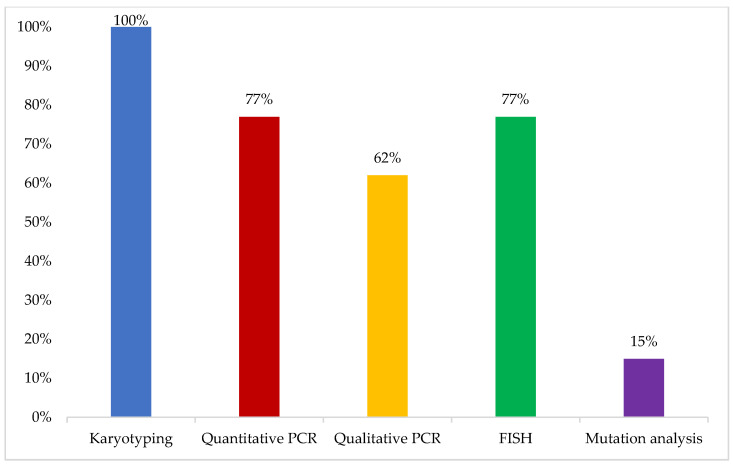
Initial test performed in patients with clinical presentation compatible with CML diagnosis. Choosing several answers was allowed. FISH: fluorescence in situ hybridization; PCR: polymerase chain reaction.

**Figure 2 cancers-16-02114-f002:**
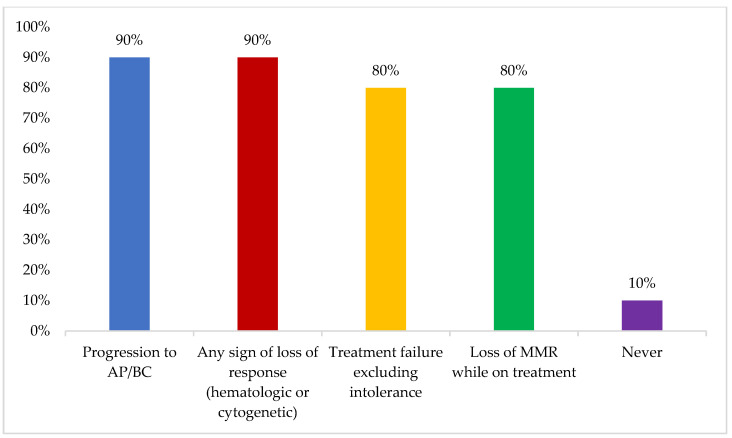
Reasons for assessing mutation status. Choosing several answers was allowed. AP/BC: accelerated phase/blast crisis; MMR: major molecular response. Other options included: “significant BCR-ABL rise from 0.001 to 0.1 (1 log increase) after achieving MMR” (0%).

**Figure 3 cancers-16-02114-f003:**
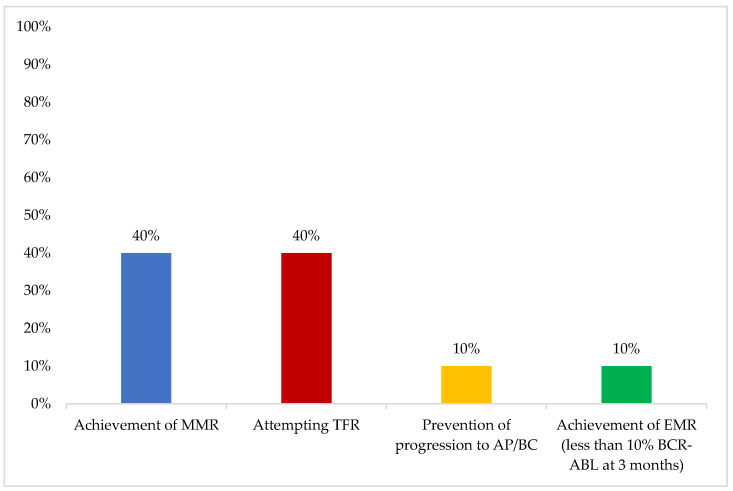
Most important first-line treatment objective in young patients. One answer only was allowed. AP/BC: AP/BC: accelerated phase/blast crisis; CCR: complete cytogenetic response; DMR: deep molecular remission; EMR: early molecular response; MMR: major molecular response; MR4: molecular response 4; TFR: treatment-free remission. Other options included: “overall survival”, “achievement of CCR”, “quality of life”, “minimize adverse events”, and “achievement of DMR (at least MR4)” (0%).

**Figure 4 cancers-16-02114-f004:**
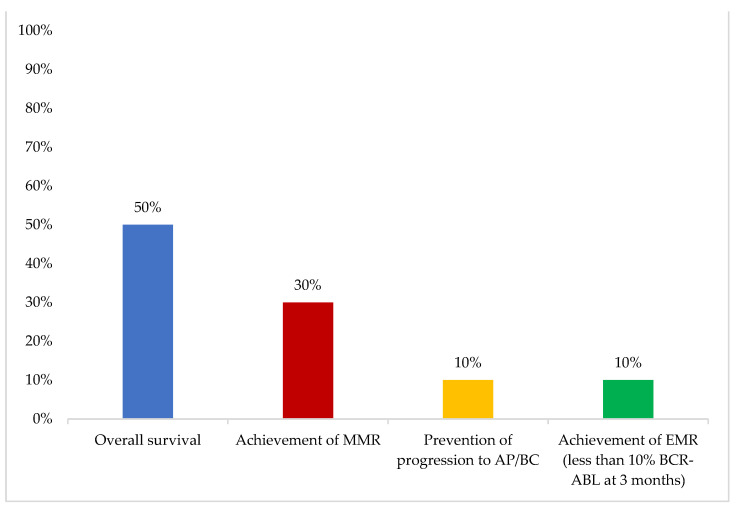
Most important first-line treatment objective in old patients. One answer only was allowed. AP/BC: AP/BC: accelerated phase/blast crisis; CCR: complete cytogenetic response; DMR: deep molecular remission; EMR: early molecular response; MMR: major molecular response; MR4: molecular response 4; TFR: treatment-free remission. Other options included: “overall survival”, “achievement of CCR”, “quality of life”, “minimize adverse events”, and “achievement of DMR (at least MR4)” (0%).

**Figure 5 cancers-16-02114-f005:**
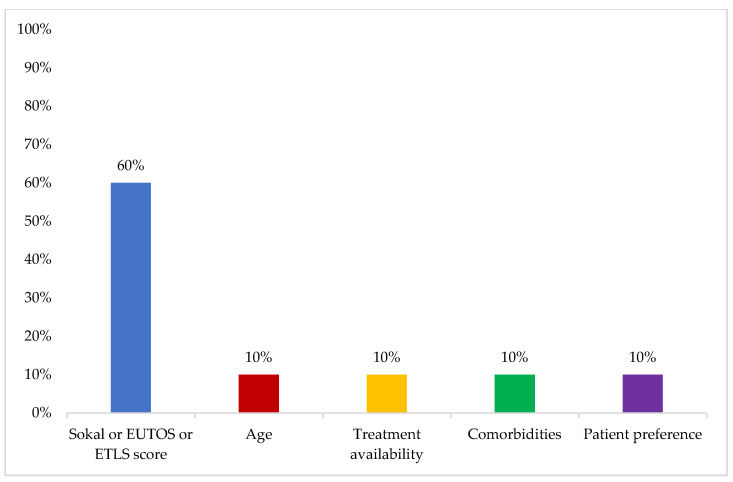
Most important factor influencing treatment decision for first-line therapy. One answer only was allowed. BID: twice daily; QD: once daily; TFR: treatment-free remission. Other options included: “dose regimen (QD/BID)” and “TFR attempt” (0%).

**Figure 6 cancers-16-02114-f006:**
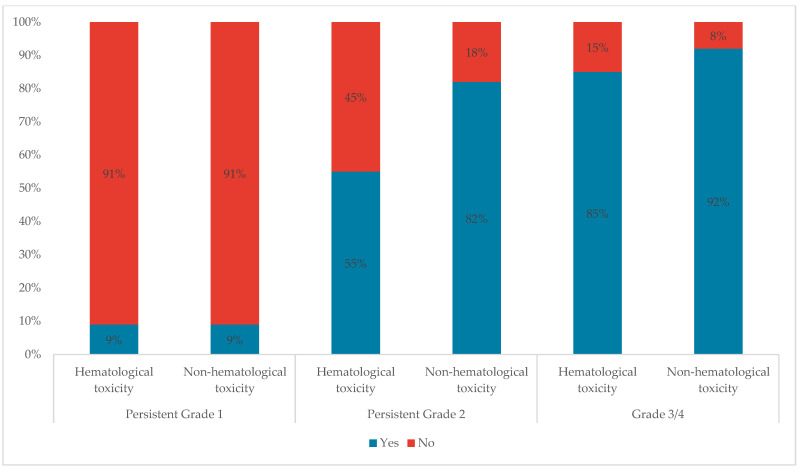
Treatment change for different grades of hematological and non-hematological toxicities.

**Table 1 cancers-16-02114-t001:** Percentage of patients on different treatments according to line of treatment.

	Imatinib	Imatinib Generic	Dasatinib	Nilotinib	Ponatinib	Bosutinib	Other Treatment
First-line (63%)	32%	23%	24%	17%	1%	4%	0%
Second-line (26%)	7%	1%	44%	36%	7%	5%	1%
Third-line (11%)	4%	1%	30%	28%	27%	11%	0%

## Data Availability

All data are contained within the article.

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
