# Peer review of "Current Status and Management of Chronic Myeloid Leukemia in the Gulf Region: Survey Results and Expert Opinion"

_cancers, 2024, doi:10.3390/cancers16112114_

Round 1

Reviewer 1 Report

Comments and Suggestions for Authors

In this paper, Saglio et al. described the incidence and the management of chronic myeloid leukemia (CML) in the Gulf region. The paper is the result of the 15 expert meeting. The paper is well written.

Nevertheless, the paper is too long and needs a major reduction (introduction and discussion). The authors must pay attention to the figure’s presentations. the figures need to be presented better by removing the negative values and being a bit more careful with the presentation. 

The nil value must be removed in the figure and can be added to the legend. The conclusion can be opened to a new horizons in the personalized management of CML patient in Gulf region.

Comments on the Quality of English Language

The quality of English is well

Reviewer 2 Report

Comments and Suggestions for Authors

Date: 18/03/2024

COMMENTS

The paper titled "Current Status and Management of Chronic Myeloid Leukemia 2 in the Gulf Region: Survey Results and Expert Opinion" is a captivating piece of interesting servey research that falls within the purview of Journal of Cancer. The article can be interesting after addressing following comments.

1.      In material and Methods section line no 98 add the complete name and affiliation of the moderator. Also add the name, affiliation of the invited speaker with their title talk.

2.      Add the questionnaire for the pre-meeting survey was sent to all participants before the meeting.

3.      Specify the recent guidelines for the use of treatments beyond second-line in CML treatment.

4.      What was the criteria for second line and third line treatment and its limitation must add in the material and methods section. Also add the list of drug used in different line treatments.

5.      Why the authors selected the very less number of participant for this survey while GCC having sufficient participant? What are the criteria for the selection of participant for this survey also mentioned in the material and methods

6.      Where this study was carried out by different 15 participant and they have taken ethical approval for the sharing this information?

7.      What was the treatment objectives and its criteria? Add in the material and methods.

8.      What was the criteria for the Treatment-related toxicities, discontinuation must add in the material and methods.

9.      How many number of sample studied for this survey?

10.  What was the Value of TKIs at lower doses?

11.  Is authors get any post TKI resistance during the treatment?

Reviewer 3 Report

Comments and Suggestions for Authors

The manuscript presents the summary of a survey and the subsequent meeting experts conducted in 2023 to address the current status of Chronic Myeloid Leukemia (CML) management and treatment care in the Gulf region. The survey was answered by 15 CML experts, and the group concluded that the Gulf region's CML practices align with global standards and similar surveys conducted previously (references 14 and 15), despite existing some differences and nuances specific to this region. In this regard, the experts suggest further meetings for continuous improvement of CML patient care adjusting the global guidelines to the Gulf region health specificities.  

The manuscript is well organized and presented clearly and will be of great interest to CML health care providers in this region.

Comments on the Quality of English Language

The English level is generally good, and the manuscript is understandable. However, sometimes in the text the message could be clearer, and the sentences better constructed.

Reviewer 4 Report

Comments and Suggestions for Authors

The results should be better. 

Lumping sokal, EUTOS, ELTS together as a score is not helpful as in the era of TKI, ELTS is preferable. Although a lot of experts still use SOKAL. If the authors ca. Elaborate it would be good.

The reason for changing TKI is relation to toxicity was very brief. Since this is a paper meant to understand the practice in the region, it would be helpful to understand what parameters one uses to decide on change of treatment.

Rechallenge of the TKI was also not mentioned. In what circumstance would a rechallenge be considered vs changing treatment. 

Not sure if quality of life of patients was a question asked but it would be good to include if there was some information on it. 
